# Monitoring sodium content in packaged foods sold in the Americas and compliance with the updated regional sodium reduction targets

Yahan Yang[1], Nadia Flexner[1,2], Maria Victoria Tiscornia[3], Leila Guarnieri[3], Adriana Blanco-Metzler[4], Hilda Núñez-Rivas[4], Marlene Roselló-Araya[4], Paola Arévalo-Rodríguez[5], Maria Fernanda Kroker-Lobos[5], Francisco Diez-Canseco[6], Mayra Meza-Hernández[6], Kiomi Yabiku-Soto[6], Lorena Saavedra-Garcia[6], Lorena Allemandi[7], Leo Nederveen[7], Mary R. L'Abbé[1]*

1 Department of Nutritional Sciences, Temerty Faculty of Medicine, University of Toronto, Toronto, Canada, 2 Global Health Advocacy Incubator, Washington, District of Columbia, United States of America, 3 Fundación Interamericana del Corazón Argentina, Buenos Aires, Argentina, 4 Costa Rican Institute of Research and Education on Nutrition and Health (INCIENSA), Tres Ríos, Cartago, Costa Rica, 5 INCAP Research Center for the Prevention of Chronic Diseases, Institute of Nutrition of Central America and Panama, Guatemala City, Guatemala, United States of America, 6 CRONICAS Center of Excellence in Chronic Diseases, Universidad Peruana Cayetano Heredia, Lima, Peru, 7 Pan American Health Organization/World Health Organization, Washington, District of Columbia, United States of America

* mary.labbe@utoronto.ca

## Abstract

### Background

Sodium reduction is a cost-effective measure to prevent noncommunicable diseases. The World Health Organization (WHO) established a target of a 30% relative reduction in mean population intake of sodium by 2025. The Pan American Health Organization (PAHO) published sodium reduction targets (SRTs) for packaged foods in 2015, expanding and updating the targets in 2021 to help Member States with its efforts in reducing population sodium intake.

### Objective

This study examined the current sodium levels in packaged foods among five countries in the Americas and monitored cross-sectional and longitudinal compliance with the sodium targets from 2015 to 2022.

### Methods

Food labels were systematically collected from the main supermarkets in five countries in the Americas region in 2022. Sodium levels per 100g and per kcal for collected food labels in 16 PAHO categories and 75 subcategories were analyzed and compared against the updated SRTs. Further analysis of three countries that have longitudinal data for 2015–2016, 2017–2018 and 2022 was conducted to compare sodium per 100g against the 2015 SRTs.

**Data availability statement:** The branded prepackaged food composition database used in this study are owned by the University of Toronto's Food Label Information Program for Latin America and Caribbean countries (FLIP-LAC) and FLIP Canada. Data are available upon request from the L'Abbe Lab, University of Toronto via email (labbe.lab@utoronto.ca) for researchers upon request.

**Funding:** This study was funded by the Canadian Institutes of Health Research, the Pan American Health Organization, and Resolve to Save Lives. The funders had no role in the study design, data collection and analysis, decision to publish, or preparation of the manuscript.

**Competing interests:** I have read the journal's policy and the authors of this manuscript have the following competing interests: NF is a staff member of the Global Health Advocacy Incubator. LN is a staff member of the Pan American Health Organization. The authors alone are responsible for the views expressed in this publication, and they do not necessarily represent the decisions or policies of the affiliated organizations.

## Results

A total of 25,569 food items were analyzed. Overall, *'processed meat and poultry'* had the highest sodium levels, although there were large variations within categories. 47% and 45% of products met the sodium per 100g and per kcal 2022 SRTs, respectively. Peru had the highest compliance, whereas Panama had the lowest for both targets. Among Argentina, Costa Rica and Peru, the proportion of foods meeting the 2015 PAHO lower targets were 48, 53 and 61% for 2015–2016, 2017–2018 and 2022, respectively (p < 0.001).

## Conclusions

This study showed that around half of the examined foods met their respective SRTs and there have been small improvements in compliance over time. Further efforts are required to reach the WHO's global sodium reduction goal by 2025, such as implementation of mandatory SRTs and front-of-pack labelling regulations.

## Introduction

Hypertension is one of the major risk factors for cardiovascular diseases (CVDs) and it has been estimated to account for 10.8 million deaths globally in 2019 [1],[2]. The number of adults with hypertension has been increasing in the past few decades, with more than 1 billion adults affected by hypertension in 2019, globally [3]. It has been well demonstrated that excessive sodium intake is a significant causal risk factor for the development of hypertension, and reducing dietary sodium intake can have a favorable effect on the cardiovascular system [4]. The World Health Organization (WHO) published a guideline in 2012 recommending less than 5g of salt (2g of sodium) intake per day for adults [5]. However, the global average salt intake in 2019 was estimated to be more than double the recommendation at 10.8g of salt (4.3g of sodium) per day [1].

In 2013, WHO established an action plan that included reducing salt intake by 30% by 2025, and all 194 Member States agreed to it [6]. A series of interventions has been proposed, including reformulating foods to contain less sodium, establishing public food procurement policies to limit high-sodium foods, implementing mandatory front-of-package labelling to inform consumers about products high in sodium (along with other nutrients of concern), and using mass media campaigns to reduce sodium intake [7]. However, a report released in early 2023 showed that the world is off-track to achieve this global target, with only 5% of countries (Brazil, Chile, Czech Republic, Lithuania, Malaysia, Mexico, Saudi Arabia, Spain and Uruguay) implementing at least two mandatory and comprehensive sodium reduction policies and all WHO sodium-related 'best buys'[1].

To support the Americas region in accomplishing the global target, the Pan American Health Organization (PAHO) published a set of regional SRTs (2015 PAHO targets) in 2015 for 18 food categories that were commonly sold in the region [8]. In 2019, a study examined the sodium levels in packaged foods sold in 14 Latin American and Caribbean countries (LAC) according to the 2015 PAHO targets. The study found 82% compliance with the regional target and 47% compliance with the lower and stricter targets [9]. The regional target level was the maximum level (mg/100g) or upper limit, whereas food manufacturers were encouraged to reach the lower target level that is reflective of the average level in reference countries [8]. A later study published in 2021 showed that compliance with regional targets increased from 83% to 89% from 2015–2016 to 2017–2018 among LAC [10]. Therefore, based on sodium levels in packaged foods in the region, PAHO updated the targets in 2021

to establish progressive regional SRTs for 2022 and 2025 (the 2022 and 2025 PAHO targets). Additionally, it categorized foods into 16 main categories and 75 subcategories [11,12]. Similarly, WHO published global sodium benchmarks in 2021 and subsequently updated them in 2024, defined as the lowest maximum values for each subcategory based on existing national or regional targets, for sodium levels across different food categories [13,14]. However, there has been no further follow-up study monitoring sodium levels in packaged foods in the Americas since 2018. Thus, the objectives of this study were to 1) examine the current sodium levels among five countries in the Americas (Argentina, Canada, Costa Rica, Panama, and Peru), and monitor compliance with the Updated PAHO Regional SRTs for 2022; and 2) monitor the progress of sodium reduction among three LAC (Argentina, Costa Rica and Peru) that have longitudinal data.

## Methods

This cross-sectional study was conducted using data from five countries: Argentina, Canada, Costa Rica, Panama, and Peru. Data from the Nutrition Facts table (NFt) (n = 44,570) were collected in supermarkets in Argentina (n = 4,340), Costa Rica (7,402), Panama (n = 1,509) and Peru (n = 5,378) between March and August 2022. Foods were selected from one or more of the major supermarket chains from different socioeconomic status in each country, representing a comprehensive sample of the packaged foods across different socioeconomic groups. The Food Label Information Program for Latin America and Caribbean countries (FLIP-LAC) was used for data collection and analyses. FLIP-LAC is a smartphone-based technology and web database, and the methodology was developed by The University of Toronto (U of T) [15]. The Canadian data (n = 25,941) were extracted from the Food Label Information and Price (FLIP) database collected in 2020 [15].

Foods were classified into 16 major categories and 75 subcategories described in the Updated PAHO Regional SRTs [12]. The updated PAHO target food categorization was individually completed by each country team and the U of T research team validated the results and resolved discrepancies with country team members. The sodium content was standardized into mg/100g and mg/kcal where data were available. Foods excluded from the analyses included duplicated items, foods that could not be categorized under the PAHO categories (n = 19,001), and foods that have 0 kcal were additionally excluded from the mg/kcal analysis, due to mathematical restrictions (n = 387). Summary statistics were calculated for food products by food categories and by country. The sodium level was compared against the Updated PAHO Regional SRTs (Supporting information S1 Table) to determine the proportion of foods that met or exceeded the target level set for 2022.

Since three countries in this study had baseline data from 2015–2016 and 2017–2018, categorized under the 2015 PAHO targets, a further sub-analysis was conducted by categorizing the foods collected in 2022 from Argentina, Costa Rica and Peru into the 2015 PAHO target categories (18 food categories) [8]. Sodium levels were compared against the 2015 PAHO regional (upper) target level as well as the lower target level. Proportion of foods meeting the regional targets was compared among the three timepoints. Comparisons between the 2015–2016 and 2022 were analyzed using a Chi-Square test, or Fisher's exact test for cells with < 5 counts. All analyses were conducted with R studio (5.12.10) and Microsoft Excel (2016).

## Results

This study included a total of 25,569 items across 16 PAHO sodium reduction main categories and 75 subcategories (Argentina n = 2,515, Canada n = 15,268, Costa Rica n = 3,875, Panama n = 1,121, and Peru n = 2,790).

## Sodium levels per 100g and per kcal by PAHO categories

For the per 100g analysis, 23,663 foods were assessed. *'Processed meat and poultry'* and *'sauces, dips, gravy, and condiments'* had the highest median sodium levels per 100g (both at 800mg/100g), followed by *'fats and oils'* (720mg/100g) (Table 1). The variations within categories were high, particularly among *'sauces, dips, gravy and condiments'* (SD = 5,733mg/100g), *'processed fish and seafood'* (SD = 972mg/100g) and *'processed vegetables, beans and legumes'* (SD = 599mg/100g). There were variations between the median sodium levels between countries, with the largest variation within *'sauces, dips, gravy, and condiments'*, *'soy products and meat alternatives'* and *'processed meat and poultry'*. The lowest variation between countries was within *'fresh or dried plain pasta and noodles'*, *'processed fish and seafood'* and *'ready-made foods'*.

For the per kcal analysis, 23,236 foods were assessed. *'Sauces, dips, gravy, and condiments'*, *'soups'* and *'processed vegetables, beans, and legumes'* have the highest median sodium level per kcal of 7.8mg/kcal, 6.0mg/kcal and 5.1mg/kcal, respectively (Table 2). The lowest median sodium levels per kcal were among *'fresh or dried plain pasta and noodles'* (0mg/kcal), *'granola and energy bars and nut butters/spreads'* (0.5mg/kcal) and *'cakes, biscuits, pastries, and sweet breads'* (0.7mg/kcal). The largest variations between countries were found among *'sauces, dips gravy and condiments'*, *'soups'* and *'soy products and meat alternatives*. Variation within category was highest among *'sauces, dips, gravy, and condiments'* (SD = 41.4mg/kcal), *'processed vegetables, beans and legumes'* (SD = 20.2mg/kcal) and *'soups'* (SD = 17.8mg/kcal).

## Proportion of foods meeting the 2022 PAHO regional sodium targets by categories and country

Overall, 11,037 of 23,663 (47%) analyzed foods met their respective 2022 PAHO sodium targets (mg/100g). Peru and Argentina had the highest proportion of compliance (52% and 50%, respectively), whereas Panama had the lowest (36%) (Fig 1). By food category, the highest proportion of products meeting the regional target was among *'ready-made foods'* (77%), *'processed meat and poultry'* (53%) and *'cheese'* (50%). The lowest proportion of foods meeting the regional target was among *'bread products'* (30%), *'granola and energy bars and nut butters/spreads'* (37%) and *'processed fish and seafood'* (41%). The largest variation between countries was observed in *'soy products and meat alternatives'* (range: 28% to 85%) and *'soups'* (range: 31% to 54%) (Table 3).

After excluding 236 products in categories that have no available targets, 23,000 foods were included in the analysis of compliance with the PAHO SRTs for sodium levels per kcal. Overall, 10,304 of 23,000 (45%) foods met their respective sodium targets (mg/kcal). Peru had the highest proportion of compliance (52%), followed by Costa Rica (48%), whereas Panama only had 37% compliance (Table 4). By category, the highest proportion of products meeting the regional target was among *'savoury snacks'* (71%), *'soups'* (67%) and *'corn derivatives'* (67%). The lowest proportion of foods meeting the regional target was among *'bread products'* (31%), *'cheese'* (35%) and *'ready-made foods'* (35%). The largest variation between countries was observed in 'soy products and meat alternatives' (range: 33% to 80%) and 'fresh or dried plain pasta and noodles' (range: 16% to 55%) (Table 4).

## Changes in the proportion of foods meeting the old 2015 PAHO sodium targets

The longitudinal analysis included a total of 10,571 foods, of which 2,915 foods were from the 2015–2016 collection, 3,241 foods were from the 2017–2018 collection, and 4,415 foods were

**Table 1. Distribution of sodium content (mg) per 100g/ml of packaged foods per PAHO food category at the regional level and by country.**

| Category | Country | Products (n) | Mean (mg) | SD (mg) | Minimum (mg) | q25 (mg) | Median (mg) | q75 (mg) | Maximum (mg) |
|---|---|---|---|---|---|---|---|---|---|
| 1. Bread, bread products and crisp breads | **Regional** | **1564** | **504** | **309** | **0** | **364** | **467** | **600** | **5300** |
| | Argentina | 226 | 476 | 288 | 0 | 365 | 457 | 608 | 2902 |
| | Canada | 1019 | 514 | 233 | 0 | 375 | 483 | 600 | 1700 |
| | Costa Rica | 188 | 551 | 615 | 0 | 291 | 473 | 658 | 5300 |
| | Panama | 44 | 527 | 154 | 240 | 444 | 500 | 592 | 1040 |
| | Peru | 87 | 354 | 113 | 40 | 311 | 365 | 396 | 676 |
| 2. Cakes, biscuits, pastries, and sweet breads | **Regional** | **3530** | **371** | **269** | **0** | **191** | **308** | **469** | **2600** |
| | Argentina | 626 | 316 | 254 | 0 | 129 | 251 | 430 | 2110 |
| | Canada | 2028 | 400 | 280 | 0 | 224 | 326 | 500 | 2600 |
| | Costa Rica | 512 | 348 | 248 | 0 | 179 | 280 | 442 | 1214 |
| | Panama | 99 | 415 | 313 | 34 | 201 | 354 | 537 | 1667 |
| | Peru | 265 | 303 | 190 | 0 | 179 | 267 | 367 | 1063 |
| 3. Corn derivatives | **Regional** | **58** | **476** | **447** | **0** | **103** | **349** | **834** | **1600** |
| | Costa Rica | 43 | 513 | 487 | 0 | 75 | 396 | 847 | 1600 |
| | Panama | 8 | 348 | 336 | 15 | 115 | 272 | 479 | 983 |
| | Peru | 7 | 399 | 270 | 63 | 343 | 344 | 368 | 960 |
| 4. Breakfast cereal | **Regional** | **1089** | **294** | **223** | **0** | **98** | **283** | **442** | **2388** |
| | Argentina | 87 | 232 | 177 | 0 | 77 | 240 | 323 | 810 |
| | Canada | 617 | 325 | 234 | 0 | 105 | 364 | 500 | 2388 |
| | Costa Rica | 195 | 278 | 206 | 0 | 113 | 268 | 374 | 1307 |
| | Panama | 48 | 339 | 215 | 0 | 205 | 350 | 408 | 1133 |
| | Peru | 142 | 202 | 187 | 0 | 18 | 183 | 347 | 775 |
| 5. Savoury snacks | **Regional** | **1692** | **545** | **354** | **0** | **305** | **500** | **731** | **3000** |
| | Argentina | 143 | 602 | 250 | 0 | 429 | 604 | 780 | 1292 |
| | Canada | 962 | 556 | 348 | 0 | 326 | 520 | 724 | 3000 |
| | Costa Rica | 336 | 516 | 390 | 0 | 202 | 445 | 720 | 2467 |
| | Panama | 92 | 646 | 467 | 27 | 358 | 532 | 842 | 2491 |
| | Peru | 159 | 434 | 276 | 19 | 266 | 359 | 577 | 1434 |
| 6. Cheese | **Regional** | **1813** | **696** | **421** | **0** | **475** | **667** | **800** | **7143** |
| | Argentina | 256 | 642 | 425 | 0 | 346 | 577 | 877 | 3300 |
| | Canada | 1167 | 698 | 385 | 0 | 500 | 667 | 767 | 3400 |
| | Costa Rica | 214 | 717 | 584 | 0 | 506 | 641 | 889 | 7143 |
| | Panama | 68 | 898 | 423 | 32 | 634 | 780 | 1253 | 1800 |
| | Peru | 108 | 629 | 348 | 10 | 398 | 608 | 800 | 1800 |
| 7. Processed vegetables, beans, and legumes | **Regional** | **1976** | **546** | **599** | **0** | **168** | **326** | **767** | **4400** |
| | Argentina | 151 | 677 | 875 | 0 | 73 | 240 | 1148 | 2885 |
| | Canada | 1244 | 539 | 582 | 0 | 153 | 324 | 767 | 4333 |
| | Costa Rica | 394 | 519 | 508 | 0 | 220 | 346 | 640 | 3500 |
| | Panama | 109 | 521 | 657 | 0 | 232 | 280 | 460 | 4400 |
| | Peru | 78 | 577 | 536 | 4 | 200 | 360 | 929 | 2300 |
| 8. Processed meat and poultry | **Regional** | **1776** | **896** | **567** | **0** | **527** | **800** | **1036** | **5460** |
| | Argentina | 178 | 962 | 747 | 46 | 593 | 753 | 1036 | 5460 |
| | Canada | 1210 | 881 | 506 | 0 | 511 | 800 | 1026 | 3143 |
| | Costa Rica | 184 | 1071 | 759 | 5 | 722 | 908 | 1225 | 4800 |
| | Panama | 91 | 790 | 394 | 96 | 626 | 769 | 969 | 2495 |
| | Peru | 113 | 763 | 546 | 198 | 335 | 600 | 900 | 2640 |

*(Continued)*

**Table 1.** (Continued)

| Category | Country | Products (n) | Mean (mg) | SD (mg) | Minimum (mg) | q25 (mg) | Median (mg) | q75 (mg) | Maximum (mg) |
|---|---|---|---|---|---|---|---|---|---|
| 9. Processed fish and seafood | **Regional** | **946** | **514** | **972** | **0** | **255** | **354** | **487** | **9940** |
| | Argentina | 54 | 541 | 1085 | 0 | 205 | 303 | 457 | 5913 |
| | Canada | 579 | 554 | 1132 | 0 | 255 | 368 | 504 | 9940 |
| | Costa Rica | 175 | 477 | 681 | 22 | 275 | 345 | 491 | 5938 |
| | Panama | 72 | 366 | 178 | 10 | 271 | 360 | 446 | 985 |
| | Peru | 66 | 403 | 227 | 70 | 269 | 350 | 450 | 1152 |
| 10. Soy products and meat alternatives | **Regional** | **274** | **478** | **278** | **0** | **355** | **421** | **577** | **1929** |
| | Argentina | 77 | 382 | 120 | 0 | 328 | 376 | 433 | 784 |
| | Canada | 116 | 555 | 344 | 0 | 393 | 473 | 632 | 1929 |
| | Costa Rica | 50 | 547 | 236 | 5 | 400 | 539 | 678 | 1375 |
| | Panama | 5 | 204 | 276 | 6 | 12 | 24 | 360 | 620 |
| | Peru | 26 | 339 | 177 | 15 | 301 | 360 | 386 | 680 |
| 11. Soups | **Regional** | **715** | **316** | **302** | **0** | **216** | **276** | **336** | **4538** |
| | Argentina | 45 | 319 | 138 | 32 | 259 | 285 | 341 | 702 |
| | Canada | 485 | 264 | 125 | 0 | 201 | 260 | 320 | 792 |
| | Costa Rica | 83 | 356 | 358 | 18 | 221 | 309 | 351 | 3250 |
| | Panama | 48 | 303 | 282 | 62 | 188 | 287 | 323 | 1797 |
| | Peru | 54 | 727 | 776 | 97 | 328 | 405 | 953 | 4538 |
| 12. Ready-made foods, convenience foods, and mixed dishes | **Regional** | **2133** | **412** | **283** | **0** | **254** | **341** | **486** | **3000** |
| | Argentina | 144 | 412 | 174 | 31 | 286 | 397 | 519 | 973 |
| | Canada | 1784 | 400 | 260 | 0 | 250 | 333 | 464 | 2050 |
| | Costa Rica | 116 | 493 | 398 | 0 | 160 | 421 | 726 | 1900 |
| | Panama | 24 | 497 | 322 | 0 | 309 | 440 | 749 | 1200 |
| | Peru | 65 | 570 | 589 | 85 | 317 | 380 | 530 | 3000 |
| 13. Fresh or dried plain pasta and noodles | **Regional** | **1108** | **20** | **101** | **0** | **0** | **3** | **11** | **2040** |
| | Argentina | 202 | 16 | 35 | 0 | 9 | 10 | 11 | 216 |
| | Canada | 571 | 22 | 99 | 0 | 0 | 0 | 12 | 1789 |
| | Costa Rica | 125 | 32 | 195 | 0 | 0 | 0 | 2 | 2040 |
| | Panama | 39 | 7 | 10 | 0 | 0 | 4 | 11 | 36 |
| | Peru | 171 | 12 | 62 | 0 | 0 | 2 | 5 | 770 |
| 14. Granola and energy bars and nut butters/ spreads | **Regional** | **967** | **222** | **146** | **0** | **115** | **221** | **313** | **893** |
| | Argentina | 33 | 137 | 119 | 0 | 58 | 104 | 208 | 429 |
| | Canada | 736 | 235 | 140 | 0 | 136 | 232 | 320 | 867 |
| | Costa Rica | 117 | 188 | 170 | 0 | 60 | 182 | 280 | 893 |
| | Panama | 31 | 285 | 138 | 56 | 156 | 281 | 391 | 500 |
| | Peru | 50 | 138 | 122 | 0 | 34 | 83 | 280 | 391 |
| 15. Fats and oils | **Regional** | **1044** | **718** | **435** | **0** | **567** | **720** | **867** | **11000** |
| | Argentina | 86 | 542 | 472 | 0 | 110 | 629 | 806 | 2520 |
| | Canada | 692 | 730 | 243 | 0 | 600 | 733 | 867 | 1800 |
| | Costa Rica | 144 | 736 | 929 | 0 | 415 | 673 | 900 | 11000 |
| | Panama | 48 | 835 | 235 | 383 | 692 | 817 | 972 | 1867 |
| | Peru | 74 | 708 | 334 | 0 | 400 | 700 | 884 | 1571 |

*(Continued)*

**Table 1.**  (Continued)

| Category | Country | Products (n) | Mean (mg) | SD (mg) | Minimum (mg) | q25 (mg) | Median (mg) | q75 (mg) | Maximum (mg) |
|---|---|---|---|---|---|---|---|---|---|
| 16. Sauces, dips, gravy and condiments | **Regional** | **2978** | **3157** | **5733** | **0** | **400** | **800** | **2800** | **36000** |
| | Argentina | 203 | 1946 | 3974 | 0 | 275 | 483 | 1283 | 24000 |
| | Canada | 2058 | 3149 | 5493 | 0 | 417 | 833 | 3200 | 36000 |
| | Costa Rica | 341 | 3222 | 6367 | 0 | 385 | 824 | 2333 | 32750 |
| | Panama | 153 | 4533 | 7137 | 0 | 528 | 1110 | 4550 | 34667 |
| | Peru | 223 | 3290 | 6874 | 0 | 362 | 681 | 1413 | 32660 |
| Total | **Regional** | 23663 | 820 | 2261 | 0 | 233 | 402 | 714 | 36000 |
| | Argentina | 2511 | 570 | 1286 | 0 | 152 | 360 | 640 | 24000 |
| | Canada | 15268 | 853 | 2251 | 0 | 256 | 429 | 733 | 36000 |
| | Costa Rica | 3217 | 782 | 2292 | 0 | 200 | 393 | 721 | 32750 |
| | Panama | 979 | 1153 | 3194 | 0 | 269 | 460 | 829 | 34667 |
| | Peru | 1688 | 781 | 2705 | 0 | 172 | 350 | 610 | 32660 |

from the 2022 collection (countries: Argentina, Costa Rica, and Peru), covering 18 PAHO sodium food categories (2015) (Fig 2). Overall, the proportion of products meeting the 2015 PAHO SRTs across the three countries increased significantly from 2015–2016 (82%) to 2022 (90%) (Fig 2A, p < 0.001). Similarly, the proportion of products meeting the 2015 PAHO lower targets was 48%, 53% and 61% for 2015–2016, 2017–2018 and 2022, respectively (Fig 2B, p < 0.001). There was also a significant increase in the proportion of products meeting the regional targets, with increases of 9%, 9% and 5% from 2015–2022 for Argentina, Costa Rica, and Peru, respectively (Fig 3A). Additionally, there was a 16%, 11% and 14% increase in the proportion of products meeting the lower targets from 2015 to 2022 for Argentina, Costa Rica, and Peru, respectively (Fig 3B). A further breakdown by food categories is presented in Supporting information S2 Table.

## Discussion

This study provides an up-to-date assessment of sodium levels in packaged foods sold in five countries in the Americas, following the publication of the Updated PAHO Regional SRTs [12]. Our results show important variation in sodium levels among key food categories, and nearly half of the assessed products complied with 2022 Updated PAHO Regional SRTs. Additionally, when assessing compliance with previous 2015 PAHO SRTs, our results indicate that the proportion of foods meeting these targets has gradually increased from 82% to 90%. However, progress has been slow, and further improvements are needed to achieve compliance with the Updated PAHO Regional SRTs. Accelerated actions are essential to achieve sodium reduction goals. These data shed light on the progress in different countries in the region towards the current sodium reduction interventions, aligned with the WHO's SHAKE technical package for sodium reduction [16] and other initiatives aimed at achieving the 2025 global target of a 30% relative reduction in mean population sodium intake.

The results show that '*processed meat and poultry*' and '*sauces, dips, gravy, and condiments*' food categories have the highest median sodium levels per 100g. This finding is consistent with a previous regional study including 14 LAC [9], which reported a similar median sodium level for '*processed meats*' (870mg/100g). Similarly, another regional study, including 4 LAC also showed that the highest sodium level was reported in '*bouillon cubes and powders*', '*meat and fish seasonings*' and '*cured and preserved meats*' [10]. In addition to the sodium levels by

**Table 2. Distribution of sodium content (mg) per kcal of packaged foods, by PAHO category at the regional level and by country.**

| Category | Country | Prod-uct (n) | Mean (mg) | SD (mg) | Minimum (mg) | q25 (mg) | Median (mg) | q75 (mg) | Maximum (mg) |
|---|---|---|---|---|---|---|---|---|---|
| 1. Bread, bread products and crisp breads | **Regional** | **1556** | **1.7** | **1.0** | **0.0** | **1.2** | **1.6** | **2.0** | **17.7** |
| | Argentina | 226 | 1.6 | 0.8 | 0.0 | 1.3 | 1.7 | 2.0 | 5.0 |
| | Canada | 1011 | 1.7 | 0.7 | 0.0 | 1.3 | 1.6 | 2.0 | 5.0 |
| | Costa Rica | 187 | 1.8 | 2.0 | 0.0 | 0.9 | 1.6 | 2.2 | 17.7 |
| | Panama | 45 | 1.8 | 0.7 | 0.8 | 1.5 | 1.7 | 2.1 | 4.7 |
| | Peru | 87 | 1.2 | 0.5 | 0.1 | 0.9 | 1.3 | 1.5 | 2.4 |
| 2. Cakes, biscuits, pastries, and sweet breads | **Regional** | **3524** | **0.9** | **0.8** | **0.0** | **0.5** | **0.7** | **1.3** | **23.0** |
| | Argentina | 625 | 0.8 | 0.6 | 0.0 | 0.3 | 0.6 | 1.1 | 4.6 |
| | Canada | 2024 | 1.0 | 0.9 | 0.0 | 0.5 | 0.8 | 1.4 | 23.0 |
| | Costa Rica | 511 | 0.9 | 0.7 | 0.0 | 0.4 | 0.7 | 1.2 | 3.5 |
| | Panama | 99 | 1.0 | 0.9 | 0.1 | 0.5 | 0.8 | 1.3 | 5.0 |
| | Peru | 265 | 0.7 | 0.4 | 0.0 | 0.4 | 0.6 | 0.9 | 2.3 |
| 3. Corn derivatives | **Regional** | **58** | **1.3** | **1.2** | **0.0** | **0.5** | **1.0** | **2.0** | **6.3** |
| | Costa Rica | 43 | 1.2 | 1.0 | 0.0 | 0.2 | 0.8 | 2.0 | 3.2 |
| | Panama | 8 | 2.0 | 2.1 | 0.1 | 0.5 | 1.3 | 2.7 | 6.3 |
| | Peru | 7 | 1.1 | 0.4 | 0.9 | 0.9 | 1.0 | 1.0 | 2.1 |
| 4. Breakfast cereal | **Regional** | **1090** | **0.8** | **0.6** | **0.0** | **0.3** | **0.8** | **1.2** | **6.8** |
| | Argentina | 87 | 0.6 | 0.5 | 0.0 | 0.2 | 0.6 | 0.9 | 2.2 |
| | Canada | 617 | 0.9 | 0.6 | 0.0 | 0.3 | 1.0 | 1.3 | 6.8 |
| | Costa Rica | 195 | 0.7 | 0.5 | 0.0 | 0.3 | 0.7 | 1.1 | 2.5 |
| | Panama | 49 | 0.9 | 0.6 | 0.0 | 0.5 | 0.9 | 1.1 | 3.1 |
| | Peru | 142 | 0.5 | 0.5 | 0.0 | 0.0 | 0.5 | 0.9 | 2.1 |
| 5. Savoury snacks | **Regional** | **1691** | **1.1** | **0.8** | **0.0** | **0.6** | **1.0** | **1.4** | **6.7** |
| | Argentina | 143 | 1.2 | 0.6 | 0.0 | 0.8 | 1.2 | 1.7 | 2.9 |
| | Canada | 960 | 1.1 | 0.7 | 0.0 | 0.6 | 1.0 | 1.4 | 6.7 |
| | Costa Rica | 336 | 1.0 | 0.9 | 0.0 | 0.4 | 0.8 | 1.4 | 5.4 |
| | Panama | 93 | 1.2 | 0.9 | 0.0 | 0.7 | 1.0 | 1.6 | 4.4 |
| | Peru | 159 | 0.9 | 0.6 | 0.0 | 0.5 | 0.7 | 1.1 | 3.2 |
| 6. Cheese | **Regional** | **1807** | **2.3** | **1.5** | **0.0** | **1.5** | **1.9** | **2.6** | **25.0** |
| | Argentina | 256 | 2.1 | 1.3 | 0.0 | 1.3 | 1.9 | 2.6 | 9.7 |
| | Canada | 1162 | 2.3 | 1.3 | 0.0 | 1.6 | 1.9 | 2.5 | 10.3 |
| | Costa Rica | 214 | 2.4 | 2.4 | 0.0 | 1.5 | 1.9 | 2.7 | 25.0 |
| | Panama | 68 | 3.0 | 1.7 | 0.2 | 1.7 | 2.4 | 4.1 | 7.5 |
| | Peru | 107 | 2.1 | 1.2 | 0.0 | 1.3 | 1.8 | 2.8 | 6.4 |
| 7. Processed vegetables, beans, and legumes | **Regional** | **1933** | **11.6** | **20.2** | **0.0** | **2.3** | **5.1** | **10.6** | **192.0** |
| | Argentina | 148 | 9.4 | 15.0 | 0.0 | 1.3 | 3.9 | 12.4 | 123.1 |
| | Canada | 1221 | 12.7 | 22.1 | 0.0 | 2.2 | 4.9 | 10.7 | 192.0 |
| | Costa Rica | 376 | 8.6 | 11.8 | 0.0 | 3.2 | 5.3 | 9.3 | 114.3 |
| | Panama | 110 | 11.7 | 27.5 | 0.0 | 3.2 | 5.8 | 9.0 | 186.7 |
| | Peru | 78 | 12.1 | 14.4 | 0.1 | 3.0 | 6.3 | 17.4 | 84.6 |
| 8. Processed meat and poultry | **Regional** | **1772** | **4.3** | **2.8** | **0.0** | **2.4** | **3.6** | **5.6** | **23.7** |
| | Argentina | 178 | 4.6 | 3.7 | 0.3 | 2.2 | 3.4 | 5.5 | 23.7 |
| | Canada | 1204 | 4.2 | 2.4 | 0.0 | 2.3 | 3.6 | 5.5 | 12.4 |
| | Costa Rica | 184 | 5.3 | 3.8 | 0.0 | 3.1 | 4.2 | 7.0 | 23.4 |
| | Panama | 93 | 4.5 | 2.6 | 0.4 | 2.9 | 4.2 | 5.8 | 12.6 |
| | Peru | 113 | 4.0 | 2.8 | 0.8 | 2.0 | 3.3 | 4.6 | 14.6 |

*(Continued)*

**Table 2.** (Continued)

| Category | Country | Prod-uct (n) | Mean (mg) | SD (mg) | Minimum (mg) | q25 (mg) | Median (mg) | q75 (mg) | Maximum (mg) |
|---|---|---|---|---|---|---|---|---|---|
| 9. Processed fish and seafood | **Regional** | **945** | **3.5** | **7.1** | **0.0** | **1.4** | **2.2** | **3.5** | **76.5** |
| | Argentina | 54 | 3.5 | 6.4 | 0.0 | 1.3 | 2.2 | 3.0 | 40.3 |
| | Canada | 579 | 4.0 | 8.6 | 0.0 | 1.5 | 2.4 | 3.6 | 76.5 |
| | Costa Rica | 175 | 2.7 | 3.5 | 0.1 | 1.2 | 1.9 | 3.1 | 31.7 |
| | Panama | 71 | 2.9 | 1.9 | 0.5 | 1.4 | 2.4 | 3.9 | 12.3 |
| | Peru | 66 | 2.5 | 1.8 | 0.2 | 1.4 | 2.0 | 3.2 | 9.1 |
| 10. Soy products and meat alternatives | **Regional** | **272** | **2.7** | **1.5** | **0.0** | **1.8** | **2.5** | **3.4** | **8.2** |
| | Argentina | 77 | 2.2 | 1.1 | 0.0 | 1.3 | 2.1 | 2.5 | 6.5 |
| | Canada | 115 | 3.2 | 1.4 | 0.0 | 2.3 | 3.0 | 4.0 | 7.0 |
| | Costa Rica | 49 | 3.1 | 1.8 | 0.1 | 2.3 | 2.7 | 3.7 | 8.2 |
| | Panama | 5 | 1.4 | 2.3 | 0.1 | 0.1 | 0.1 | 1.3 | 5.5 |
| | Peru | 26 | 1.9 | 0.9 | 0.1 | 1.6 | 2.4 | 2.6 | 2.8 |
| 11. Soups | **Regional** | **700** | **11.7** | **17.8** | **0.0** | **3.9** | **6.0** | **11.1** | **180.0** |
| | Argentina | 45 | 10.5 | 8.0 | 2.2 | 3.7 | 9.9 | 12.1 | 40.7 |
| | Canada | 472 | 12.5 | 20.4 | 0.0 | 3.8 | 6.0 | 8.6 | 180.0 |
| | Costa Rica | 83 | 12.9 | 13.6 | 0.0 | 4.9 | 8.7 | 14.8 | 66.2 |
| | Panama | 48 | 10.7 | 6.8 | 2.0 | 4.4 | 10.8 | 16.2 | 26.5 |
| | Peru | 52 | 5.5 | 3.7 | 0.9 | 3.2 | 4.5 | 6.2 | 17.6 |
| 12. Ready-made foods, convenience foods, and mixed dishes | **Regional** | **2116** | **2.3** | **1.2** | **0.0** | **1.6** | **2.1** | **2.7** | **21.8** |
| | Argentina | 144 | 2.0 | 0.8 | 0.2 | 1.4 | 2.1 | 2.5 | 4.8 |
| | Canada | 1769 | 2.2 | 1.1 | 0.0 | 1.6 | 2.1 | 2.6 | 21.8 |
| | Costa Rica | 115 | 2.3 | 1.4 | 0.0 | 1.3 | 2.1 | 3.0 | 6.7 |
| | Panama | 24 | 2.6 | 3.2 | 0.0 | 2.0 | 2.2 | 2.5 | 16.9 |
| | Peru | 64 | 3.2 | 2.2 | 0.3 | 1.9 | 2.7 | 3.7 | 14.1 |
| 13. Fresh or dried plain pasta and noodles | **Regional** | **1110** | **0.1** | **0.3** | **0.0** | **0.0** | **0.0** | **0.0** | **5.1** |
| | Argentina | 201 | 0.0 | 0.1 | 0.0 | 0.0 | 0.0 | 0.0 | 0.8 |
| | Canada | 571 | 0.1 | 0.3 | 0.0 | 0.0 | 0.0 | 0.0 | 5.1 |
| | Costa Rica | 128 | 0.1 | 0.6 | 0.0 | 0.0 | 0.0 | 0.0 | 5.1 |
| | Panama | 39 | 0.1 | 0.2 | 0.0 | 0.0 | 0.0 | 0.0 | 0.7 |
| | Peru | 171 | 0.0 | 0.2 | 0.0 | 0.0 | 0.0 | 0.0 | 2.1 |
| 14. Granola and energy bars and nut butters/spreads | **Regional** | **967** | **0.5** | **0.4** | **0.0** | **0.3** | **0.5** | **0.8** | **4.1** |
| | Argentina | 33 | 0.3 | 0.3 | 0.0 | 0.1 | 0.3 | 0.5 | 0.9 |
| | Canada | 736 | 0.6 | 0.3 | 0.0 | 0.3 | 0.6 | 0.8 | 3.0 |
| | Costa Rica | 117 | 0.5 | 0.6 | 0.0 | 0.1 | 0.4 | 0.7 | 4.1 |
| | Panama | 32 | 0.6 | 0.2 | 0.1 | 0.4 | 0.6 | 0.8 | 0.9 |
| | Peru | 49 | 0.3 | 0.3 | 0.0 | 0.1 | 0.2 | 0.5 | 1.1 |
| 15. Fats and oils | **Regional** | **1035** | **2.6** | **3.9** | **0.0** | **1.0** | **1.8** | **3.1** | **44.0** |
| | Argentina | 85 | 2.0 | 2.0 | 0.0 | 0.5 | 1.6 | 3.0 | 10.5 |
| | Canada | 691 | 2.6 | 3.7 | 0.0 | 1.0 | 1.8 | 3.1 | 36.0 |
| | Costa Rica | 140 | 3.4 | 5.7 | 0.0 | 1.1 | 2.0 | 3.3 | 44.0 |
| | Panama | 45 | 2.9 | 4.3 | 0.6 | 1.2 | 2.0 | 3.3 | 28.0 |
| | Peru | 74 | 2.7 | 3.3 | 0.0 | 1.0 | 1.6 | 2.7 | 22.7 |

*(Continued)*

**Table 2.** (Continued)

| Category | Country | Product (n) | Mean (mg) | SD (mg) | Minimum (mg) | q25 (mg) | Median (mg) | q75 (mg) | Maximum (mg) |
|---|---|---|---|---|---|---|---|---|---|
| 16. Sauces, dips, gravy and condiments | **Regional** | **2660** | **22.0** | **41.4** | **0.0** | **3.7** | **7.8** | **19.0** | **653.2** |
| | Argentina | 190 | 28.8 | 52.2 | 0.0 | 3.0 | 8.5 | 36.0 | 455.0 |
| | Canada | 1819 | 18.3 | 30.2 | 0.0 | 3.8 | 7.7 | 18.0 | 334.7 |
| | Costa Rica | 291 | 23.6 | 42.9 | 0.0 | 2.9 | 8.0 | 22.2 | 335.0 |
| | Panama | 153 | 45.9 | 70.1 | 0.0 | 5.5 | 14.2 | 50.0 | 352.5 |
| | Peru | 207 | 28.0 | 68.8 | 0.0 | 2.6 | 6.3 | 16.2 | 653.2 |
| Total | **Regional** | **23236** | **5.2** | **17.0** | **0.0** | **0.8** | **1.7** | **3.5** | **653.2** |
| | Argentina | 2492 | 4.3 | 16.7 | 0.0 | 0.5 | 1.4 | 2.5 | 455.0 |
| | Canada | 14951 | 5.1 | 14.3 | 0.0 | 0.9 | 1.8 | 3.6 | 334.7 |
| | Costa Rica | 3144 | 4.9 | 15.5 | 0.0 | 0.6 | 1.6 | 3.6 | 335.0 |
| | Panama | 982 | 10.5 | 33.2 | 0.0 | 0.9 | 2.3 | 5.5 | 352.5 |
| | Peru | 1667 | 5.3 | 26.0 | 0.0 | 0.5 | 1.1 | 2.8 | 653.2 |

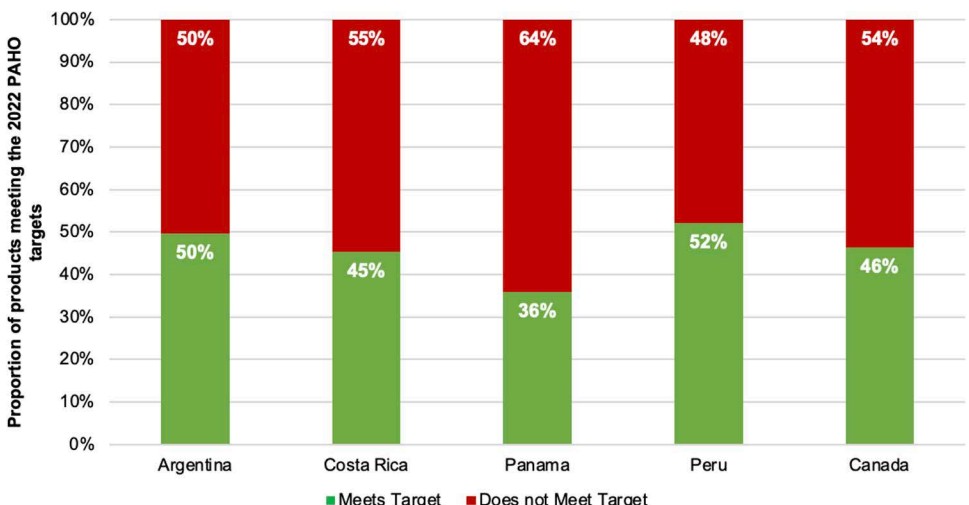

**Fig 1. Proportion of products meeting the updated 2022 PAHO Sodium Targets – by country.**

weight, sodium density (mg/kcal) accounts for variations in energy consumption and has been shown to have a stronger association with blood pressure than total intake [17]. Our results indicate that *'sauces, dips, gravy, and condiments'*, *'soups'* and *'processed vegetables, beans, and legumes'* had the highest sodium density. These results are expected, as these food categories generally have lower energy density.

Overall, the results also revealed large variations within food categories, suggesting that lower sodium options are available and demonstrating the technical feasibility of reducing sodium levels in these categories. This is important, as it highlights the importance of combining other strategies, such as front-of-pack labels to help consumers identify lower sodium options, social marketing campaigns to encourage consumers to select lower sodium foods, marketing restrictions for products with high sodium content, and nutrition standards for foods and beverages available in schools and other settings, to disincentivize the consumption of such products. However, the high average and median sodium levels in many categories

**Table 3. Proportion of products meeting the 2022 PAHO Sodium Reduction Targets** (mg per 100g/mL)**, at the regional and country level, by food category.**

| PAHO 2021 Major and Subcategory | Regional | | Argentina | | Costa Rica | | Panama | | Peru | | Canada | |
|---|---|---|---|---|---|---|---|---|---|---|---|---|
| | n | % (n) 2022 | n | % (n) 2022 | n | % (n) 2022 | n | % (n) 2022 | n | % (n) 2022 | n | % (n) 2022 |
| 1. Bread, bread products and crisp breads | 1564 | 30 (476) | 226 | 27 (60) | 188 | 36 (68) | 44 | 20 (9) | 87 | 55 (48) | 1019 | 29 (291) |
| 2. Cakes, biscuits, pastries and sweet breads | 3530 | 43 (1526) | 626 | 56 (351) | 512 | 42 (217) | 99 | 34 (34) | 265 | 51 (134) | 2028 | 39 (790) |
| 3. Corn derivatives | 58 | 47 (27) | 0 | NA | 43 | 44 (19) | 8 | 25 (2) | 7 | 86 (6) | 0 | NA |
| 4. Breakfast cereal | 1089 | 46 (499) | 87 | 56 (49) | 195 | 49 (96) | 48 | 38 (18) | 142 | 56 (79) | 617 | 42 (257) |
| 5. Savoury snacks | 1692 | 45 (754) | 143 | 23 (33) | 336 | 52 (174) | 92 | 36 (33) | 159 | 62 (99) | 962 | 43 (415) |
| 6. Cheese | 1813 | 50 (909) | 256 | 69 (176) | 214 | 57 (121) | 68 | 44 (30) | 108 | 60 (65) | 1167 | 44 (517) |
| 7. Processed vegetables, beans, and legumes | 1976 | 43 (852) | 151 | 41 (62) | 394 | 45 (178) | 109 | 28 (30) | 78 | 54 (42) | 1244 | 43 (540) |
| 8. Processed meat and poultry | 1776 | 53 (948) | 178 | 42 (75) | 184 | 31 (57) | 91 | 47 (43) | 113 | 65 (74) | 1210 | 58 (699) |
| 9. Processed fish and seafood | 946 | 41 (385) | 54 | 54 (29) | 175 | 42 (73) | 72 | 36 (26) | 66 | 33 (22) | 579 | 41 (235) |
| 10. Soy products and meat alternatives | 274 | 46 (127) | 77 | 69 (53) | 50 | 28 (14) | 5 | 80 (4) | 26 | 85 (22) | 116 | 29 (34) |
| 11. Soups | 715 | 48 (346) | 45 | 53 (24) | 83 | 40 (33) | 48 | 40 (19) | 54 | 15 (8) | 485 | 54 (262) |
| 12. Ready-made foods, convenience foods, and mixed dishes | 2133 | 77 (1638) | 144 | 81 (116) | 116 | 68 (79) | 24 | 67 (16) | 65 | 82 (53) | 1784 | 77 (1374) |
| 13. Fresh or dried plain pasta and noodles | 1108 | 43 (479) | 202 | 16 (33) | 125 | 57 (71) | 39 | 41 (16) | 171 | 41 (70) | 571 | 51 (289) |
| 14. Granola and energy bars and nut butters/spreads | 967 | 37 (357) | 33 | 67 (22) | 117 | 50 (58) | 31 | 35 (11) | 50 | 68 (34) | 736 | 32 (232) |
| 15. Fats and oils | 1044 | 40 (414) | 86 | 57 (49) | 144 | 50 (72) | 48 | 19 (9) | 74 | 47 (35) | 692 | 36 (249) |
| 16. Sauces, dips, gravy and condiments | 2978 | 44 (1300) | 203 | 57 (116) | 341 | 39 (132) | 153 | 34 (52) | 223 | 40 (89) | 2058 | 44 (911) |
| Total | 23663 | 47 (11037) | 2511 | 50 (1248) | 3217 | 45 (1462) | 979 | 35 (352) | 1688 | 52 (880) | 15268 | 46 (7095) |

Note: n, number of foods sampled and % (n) of products meeting the 2022 PAHO Sodium Reduction Targets (items were compared against the targets set for respective sub categories). NA, not applicable.

also suggested that governments need to encourage sodium reduction at the manufacturer level through food reformulation by setting mandatory SRTs for key food categories.

When examining compliance with the 2022 PAHO SRTs, almost half of the analyzed foods met their respective sodium targets (mg/100g). At the country level, compliance rate for the 2022 PAHO SRTs were similar across countries, except for Panama, which had a lower compliance rate of 36%. On examination of the longitudinal progress of Argentina, Costa Rica and Peru, there was an 8% increase in compliance when compared to the 2015 regional target (82% to 90%) and a 13% increase when compared to the 2015 lower regional target (48% to 61%). While this may indicate some progress, it remains insufficient to achieve the target of a 30% relative reduction in mean population intake of sodium by 2025. Therefore, more effective strategies, such as mandatory SRTs, are needed for better compliance with the set targets.

Furthermore, the results need careful interpretation due to the different stages at which each country is in its efforts to reduce sodium. These five countries each have national-level policies supporting sodium reduction. For example, Argentina is one of the few countries in the world to establish legal sodium maximum levels for certain food groups by passing Act 26905 in 2014, as well as educational campaigns and restaurant policies to reduce sodium intake [18]. There is also ongoing monitoring of Argentina's sodium content in foods. Studies

**Table 4. Proportion of products meeting the 2022 PAHO Sodium Reduction Targets** (mg/kcal)**, at the regional and country level, by food category.**

| PAHO 2021 Major and Subcategory | Regional | | Argentina | | Costa Rica | | Panama | | Peru | | Canada | |
|---|---|---|---|---|---|---|---|---|---|---|---|---|
| | n | % (n) 2022 | n | % (n) 2022 | n | % (n) 2022 | n | % (n) 2022 | n | % (n) 2022 | n | % (n) 2022 |
| 1. Bread, bread products and crisp breads | 1556 | 31 (487) | 226 | 27 (62) | 187 | 40 (74) | 45 | 24 (11) | 87 | 55 (48) | 1011 | 29 (292) |
| 2. Cakes, biscuits, pastries and sweet breads | 3524 | 43 (1526) | 625 | 52 (326) | 511 | 41 (212) | 99 | 38 (38) | 265 | 52 (139) | 2024 | 40 (811) |
| 3. Corn derivatives | 58 | 67 (39) | NA | NA | 43 | 65 (28) | 8 | 62 (5) | 7 | 86 (6) | NA | NA |
| 4. Breakfast cereal | 1090 | 40 (441) | 87 | 48 (42) | 195 | 43 (84) | 49 | 33 (16) | 142 | 55 (78) | 617 | 36 (221) |
| 5. Savoury snacks | 1691 | 71 (1205) | 143 | 57 (82) | 336 | 74 (248) | 93 | 66 (61) | 159 | 84 (134) | 960 | 71 (680) |
| 6. Cheese | 1807 | 35 (639) | 256 | 36 (92) | 214 | 43 (93) | 68 | 43 (29) | 107 | 41 (44) | 1162 | 33 (381) |
| 7. Processed vegetables, beans, and legumes | 1706 | 47 (795) | 128 | 47 (60) | 354 | 45 (161) | 95 | 33 (31) | 72 | 49 (35) | 1057 | 48 (508) |
| 8. Processed meat and poultry | 1772 | 39 (689) | 178 | 33 (59) | 184 | 30 (55) | 93 | 28 (26) | 113 | 38 (43) | 1204 | 42 (506) |
| 9. Processed fish and seafood | 945 | 48 (453) | 54 | 52 (28) | 175 | 57 (100) | 71 | 41 (29) | 66 | 56 (37) | 579 | 45 (259) |
| 10. Soy products and meat alternatives | 272 | 49 (133) | 77 | 73 (56) | 49 | 37 (18) | 5 | 80 (4) | 26 | 65 (17) | 115 | 33 (38) |
| 11. Soups | 700 | 67 (472) | 45 | 40 (18) | 83 | 48 (40) | 48 | 44 (21) | 52 | 67 (35) | 472 | 76 (358) |
| 12. Ready-made foods, convenience foods, and mixed dishes | 2107 | 35 (735) | 144 | 48 (69) | 115 | 47 (54) | 24 | 25 (6) | 64 | 23 (15) | 1760 | 34 (591) |
| 13. Fresh or dried plain pasta and noodles | 1110 | 43 (479) | 201 | 16 (33) | 128 | 55 (71) | 39 | 41 (16) | 171 | 41 (70) | 571 | 51 (289) |
| 14. Granola and energy bars and nut butters/spreads | 967 | 47 (455) | 33 | 76 (25) | 117 | 63 (74) | 32 | 38 (12) | 49 | 69 (34) | 736 | 42 (310) |
| 15. Fats and oils | 1035 | 43 (448) | 85 | 52 (44) | 140 | 41 (57) | 45 | 29 (13) | 74 | 39 (29) | 691 | 44 (305) |
| 16. Sauces, dips, gravy and condiments | 2660 | 49 (1308) | 190 | 53 (101) | 291 | 40 (116) | 153 | 27 (42) | 207 | 48 (100) | 1818 | 52 (948) |
| Total | 23000 | 45 (10304) | 2472 | 44 (1097) | 3122 | 48 (1485) | 967 | 37 (360) | 1661 | 52 (864) | 14777 | 44 (6497) |

Note: n, number of foods sampled and % (n) of products meeting the 2022 PAHO Sodium Reduction Targets (items were compared against the targets set for respective sub categories). NA, not applicable.

in 2017–2018 and 2022 showed that over 90% and 94% of surveyed food products, respectively, complied with the national sodium reduction law [19,20], similar to the current finding in compliance with the 2015 regional targets. However, this suggests that the current national limits are too lax compare with the 2022 PAHO SRTs and require further adjustment; and the necessity to include sodium source food groups that are currently not included, such as cheese and puff pastries [21].

While Costa Rica only has a voluntary sodium reduction strategy to reduce sodium levels in some key food categories, progress has been monitored continuously [22–24]. From 2013 to 2021, Costa Rica has a national plan for the reduction of sodium intake, and a private-public partnership was established in 2014 for voluntary national sodium targets. A study published by Vega-Solano et al. in 2019 showed 87% compliance with the voluntary national sodium targets [22]. The Development and Public Investment Plan 2019-2022 also contemplating the decrease in premature mortality rate due to non-communicable diseases (NCD) [24]. These policies may lead to the increased compliance rate with the 2015 sodium targets seen in 2022. In the case of Peru, there is no specific national sodium reduction strategy in place; however, Peru implemented 'high in' FOPL regulations in 2019 that require foods exceeding established thresholds for nutrients of concern (i.e., sugar, saturated fat, trans fat, and sodium) to display a 'high in' FOPL [25]. This law could indirectly motivate food industry to reformulate products

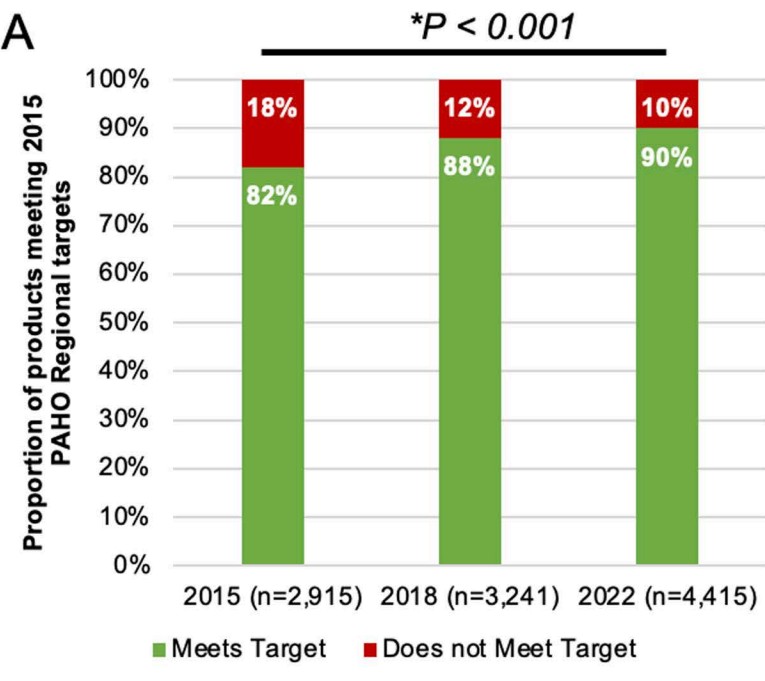

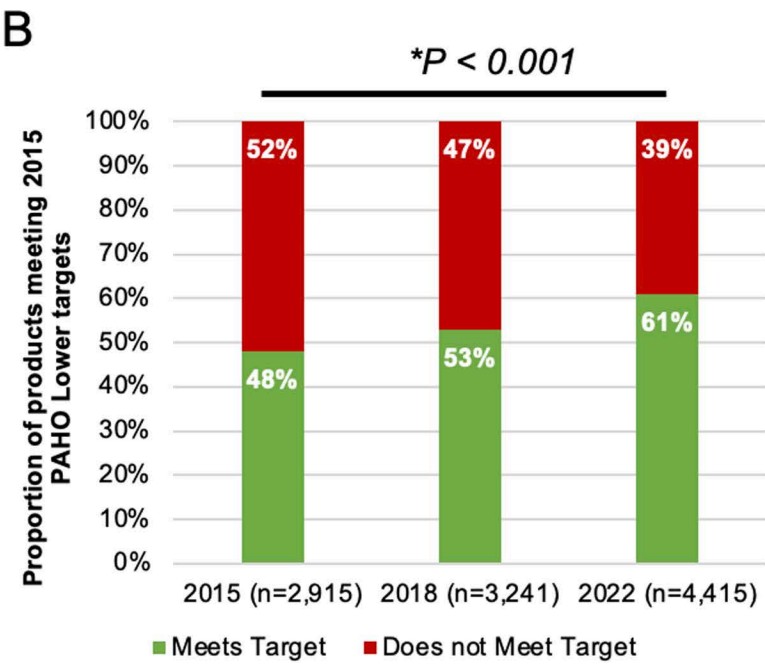

**Fig 2.** A) Proportion of products meeting the PAHO 2015 Regional Sodium Targets from 2015 to 2022 (Argentina, Costa Rica, Peru) – overall; B) Proportion of products meeting the PAHO 2015 Lower Sodium Targets from 2015 to 2022 (Argentina, Costa Rica, Peru) – overall.

in order to avoid 'high in' sodium labels, as has already been observed in Peru and other countries [26,27]. The food industry may have adjusted sodium content prior to this study, likely influenced by the implementation of the second phase of the FOPL policy. This could explain the relatively high compliance rate with the 2022 regional sodium targets in Peru and the significant increase in compliance rate when compared to the 2015 sodium targets.

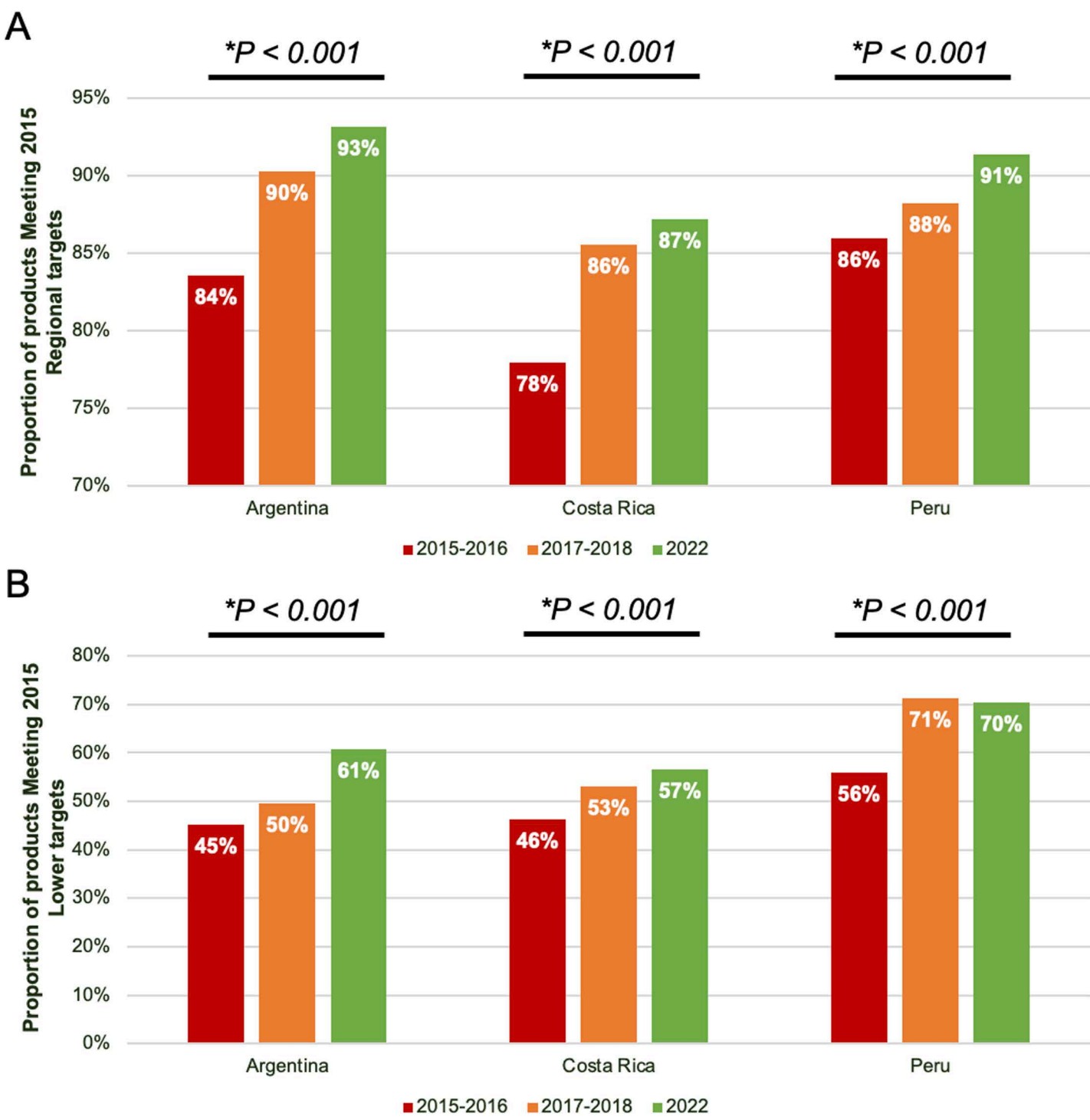

**Fig 3. A) Proportion of products meeting the 2015 PAHO Regional Sodium Targets from 2015 to 2022 – by country (2015 – 2022); B) Proportion of products meeting the 2015 PAHO Lower Sodium Targets from 2015 to 2022 – by country.**

Canada published a voluntary sodium reduction guideline for prepackaged foods in 2012 and updated the targets in 2020 [28,29]. Evaluations of this initiative have shown minimal progress over time, with only 14% of food categories meeting sodium reduction targets in

2017 [30]. In June 2022, Canada introduced its mandatory 'high in' FOPL regulations for saturated fat, sodium, and sugar in packaged foods [31]. With an enforcement date set in January 2026, follow-up study is necessary to examine the effect of this policy in Canada. On the other hand, Panama does not have a FOPL policy in place for products with critical excess nutrients, including sodium, which may explain, partially, the low compliance rate with regional SRTs. In December 2022, the 2022–2025 Action Plan for the Reduction of Sodium and Elimination of Trans Fats in Panama was launched after extensive consultations with key stakeholders from the governmental and non-governmental sectors, academia, civil society, and international cooperation agencies. The reformulation of processed foods, focusing on the reduction of sodium and trans fats, as well as the implementation of a FOPL are part of the strategies included in the Action Plan [32].

Overall, these interventions aiming to reduce sodium intake by encouraging food reformulations could have important public health impacts, as several simulation studies found that an important number of CVDs could have been prevented or delayed if the reformulation targets or sodium guidelines would have been met [33–36]. Worth noting that mandatory policies have shown to be more effective than voluntary initiatives. For instance, a recent evaluation in South Africa, one of the fewest countries with mandatory sodium levels for key food categories, showed a reduction in salt intake of 1.15 g/day during the first phase of the program (2015–2019), a significant reduction in population-level sodium intake, and achieved 75% compliance following the second, more stringent phase of legislation in 2019 [37,38], while other voluntary initiatives in countries like Brazil and Canada have shown only modest results in meeting sodium reduction targets [30,39,40]. Given the slow progress in meeting regional sodium reduction targets, our results highlight the importance of establishing and monitoring effective and feasible mandatory SRTs to prevent diet-related NCD deaths.

This study has both strengths and limitations that should be considered when interpreting our findings. Variations in sampling sizes among countries could potentially influence the overall study results. Therefore, a detailed examination of variations within each country at the subcategory level would be necessary, although out of the scope of this analysis. However, we do provide results of progress relative to the PAHO targets, at the major category level for each country. Additionally, while we selected packaged products from major supermarket chains, we recognize that we may not have captured all of the available packaged food products in the region, or fully represent all components of the population's diet. Similarly, we were unable to account for consumption patterns, and further field research is needed to better understand all the sources of sodium intake of populations. Furthermore, we were unable to conduct laboratory analyses to validate the sodium concentration in all food products, as this was beyond the scope of the current analysis. Instead, we relied on the sodium values reported by food companies, as extracted from the NFt. Although Argentina and Canada have a permitted variation margin of 20% when reporting nutrient values in NFts [41,42], the accuracy could not be validated in this study. Additionally, several products per country were excluded due to the absence of sodium level declarations on food labels, as some countries within the region lack mandatory nutrient values declaration. Regional comparison at different time points was limited to only three countries, and missing data were excluded. Including other countries and matching products in future analyses will provide a more comprehensive assessment at the regional level. However, this regional analysis benefited from the utilization of common methodologies across countries (e.g., data collection, data cleaning, and analysis). Additionally, several quality assurance measures were performed including validation of food categorization and outlier checks. Furthermore, prices of products were not included in this study. To ensure the decrease in sodium content is not driven by the lower sodium products

that might be higher in price (which likely will not target the majority of population), future studies should also consider including sales-weighted average sodium levels to account for the actual sales of foods sampled.

## Conclusion

Overall, this study provides an ongoing surveillance of the sodium content in package foods sold in five countries in the Americas, under key food categories of the Updated PAHO Regional SRTs. Around half of the examined foods met their respective SRTs and there has been some improvement in the compliance overtime, although the reductions have been modest at best. Further efforts are required to reach the WHO's global sodium reduction goal by 2025, such as implementation of national mandatory SRTs, reformulation guidelines, FOPL regulations, marketing restrictions, social marketing campaigns, and nutrition standards in schools and other settings, among others.

## Acknowledgments

The authors would like to express their gratitude to Alyssa Schermel for her support to country teams with the FLIP-LAC application and platform. Also, to country team members Luciana Castronuovo (Argentina) and Ana Atencio (Panama) for their contributions throughout the project.

## Author contributions

**Conceptualization:** Yahan Yang, Nadia Flexner, Maria Victoria Tiscornia, Leila Guarnieri, Adriana Blanco-Metzler, Hilda Núñez-Rivas, Marlene Roselló-Araya, Paola Arévalo-Rodríguez, Maria Fernanda Kroker-Lobos, Francisco Diez-Canseco, Mayra Meza-Hernández, Kiomi Yabiku-Soto, Lorena Saavedra-Garcia, Lorena Allemandi, Leo Nederveen.

**Data curation:** Yahan Yang, Nadia Flexner, Maria Victoria Tiscornia, Leila Guarnieri, Adriana Blanco-Metzler, Hilda Núñez-Rivas, Marlene Roselló-Araya, Paola Arévalo-Rodríguez, Maria Fernanda Kroker-Lobos, Francisco Diez-Canseco, Mayra Meza-Hernández, Kiomi Yabiku-Soto, Lorena Saavedra-Garcia, Mary R. L'Abbé.

**Formal analysis:** Yahan Yang, Maria Victoria Tiscornia, Leila Guarnieri, Adriana Blanco-Metzler, Hilda Núñez-Rivas, Marlene Roselló-Araya, Paola Arévalo-Rodríguez, Maria Fernanda Kroker-Lobos, Francisco Diez-Canseco, Mayra Meza-Hernández, Kiomi Yabiku-Soto, Lorena Saavedra-Garcia.

**Funding acquisition:** Lorena Allemandi, Leo Nederveen, Mary R. L'Abbé.

**Investigation:** Nadia Flexner, Lorena Allemandi, Leo Nederveen, Mary R. L'Abbé.

**Methodology:** Yahan Yang, Nadia Flexner, Lorena Allemandi, Leo Nederveen, Mary R. L'Abbé.

**Project administration:** Yahan Yang, Maria Victoria Tiscornia, Leila Guarnieri, Adriana Blanco-Metzler, Hilda Núñez-Rivas, Marlene Roselló-Araya, Paola Arévalo-Rodríguez, Maria Fernanda Kroker-Lobos, Francisco Diez-Canseco, Mayra Meza-Hernández, Kiomi Yabiku-Soto, Lorena Saavedra-Garcia, Lorena Allemandi, Leo Nederveen, Mary R. L'Abbé.

**Resources:** Lorena Allemandi, Leo Nederveen, Mary R. L'Abbé.

**Software:** Yahan Yang.

**Supervision:** Mary R. L'Abbé.

**Validation:** Yahan Yang.

**Visualization:** Yahan Yang.

**Writing – original draft:** Yahan Yang, Nadia Flexner.

**Writing – review & editing:** Yahan Yang, Nadia Flexner, Maria Victoria Tiscornia, Leila Guarnieri, Adriana Blanco-Metzler, Hilda Núñez-Rivas, Marlene Roselló-Araya, Paola Arévalo-Rodríguez, Maria Fernanda Kroker-Lobos, Francisco Diez-Canseco, Mayra Meza-Hernández, Kiomi Yabiku-Soto, Lorena Saavedra-Garcia, Lorena Allemandi, Leo Nederveen, Mary R. L'Abbé.

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
