## [Decision Letter · Decision Letter 0]

7 Jul 2024

PONE-D-24-19881Monitoring sodium content in packaged foods sold in the Americas and compliance with the Updated Regional Sodium Reduction TargetsPLOS ONE

Dear Dr. L’Abbé,

Thank you for submitting your manuscript to PLOS ONE. After careful consideration, we feel that it has merit but does not fully meet PLOS ONE’s publication criteria as it currently stands. Therefore, we invite you to submit a revised version of the manuscript that addresses the points raised during the review process.

We look forward to receiving your revised manuscript.

Kind regards,

Trung Quang Nguyen

Academic Editor

PLOS ONE

Journal Requirements:

2. Thank you for stating the following financial disclosure: "Canadian Institutes of Health Research, Pan Amercian Health Organization and Resolve to Save Lifes."

4. We note you have included a table to which you do not refer in the text of your manuscript. Please ensure that you refer to Table 5 in your text; if accepted, production will need this reference to link the reader to the Table.

Reviewers' comments:

Reviewer's Responses to Questions

**Comments to the Author**

1. Is the manuscript technically sound, and do the data support the conclusions?

Reviewer #1: Yes

Reviewer #2: Yes

Reviewer #3: Yes

2. Has the statistical analysis been performed appropriately and rigorously? 

Reviewer #1: No

Reviewer #2: Yes

Reviewer #3: No

3. Have the authors made all data underlying the findings in their manuscript fully available?

Reviewer #1: Yes

Reviewer #2: Yes

Reviewer #3: Yes

4. Is the manuscript presented in an intelligible fashion and written in standard English?

Reviewer #1: Yes

Reviewer #2: Yes

Reviewer #3: Yes

5. Review Comments to the Author

Reviewer #1: The manuscript is well structured, and the appendices and supplementary files support the findings. However, there are certain things the manuscript need to improve.

1. For the three countries where longitudinal data is available please include results for the test of significance as well if possible.

2. Add a few paragraphs to support your argument that the actual sodium concentration in the packaged foods are not different from as mentioned on the nutrition facts labels in the results section. Or mention the validation process that were used to verify how accurately the labels represent true value.

2. The authors have mentioned Atwater validation in the last paragraph of discussion line-288. Please explain how its related to validating sodium concentration, if the validation was used, please provide results accordingly.

Reviewer #2: Dear Author,

Thank you for your excellent work. I found the detail and really enjoyed in each section of the work. The title is well descriptive and abstract is well written. Introduction, Methedology and results are very well written. My concern is the discussion section is very shallow and please discuss in detail by comparing with the prevous published similar study elswere. Therefore I suggested for minor revision.

Reviewer #3: General comment:

The manuscript presents important data on sodium content in packaged foods but faces several scientific, writing, and methodological issues. Addressing these problems can significantly enhance the manuscript's clarity, reliability, and overall impact.

Specific comments:

- We don't know why some countries (like Peru and Panama) have different compliance rates. The study's conclusions may be oversimplified without examining socio-economic, regulatory and cultural differences.

- There is sampling bias. The study relies on food labels collected from supermarkets, which may not be representative of all packaged foods available in the region.

- The study uses both cross-sectional data from 2022 and longitudinal data from 2015-2022. However, the rationale for combining these two approaches is not fully justified, which may affect the consistency and comparability of the results.

- The study relies on food label information, which may not always accurately reflect the actual sodium content due to possible differences in labelling practices. This limitation is not adequately addressed.

- The manuscript does not provide details on how supermarkets were selected for food label collection, raising concerns about potential selection bias.

- The manuscript does not address how missing data were handled, which can be a critical issue in longitudinal studies.

- There is little discussion on potential confounding variables that could affect sodium content in foods, such as changes in food formulation practices, consumer preferences, or regulatory changes over time.

6. PLOS authors have the option to publish the peer review history of their article (what does this mean? ). If published, this will include your full peer review and any attached files.

**Do you want your identity to be public for this peer review?** For information about this choice, including consent withdrawal, please see our Privacy Policy .

Reviewer #1: **Yes: ** Saroj Adhikari

Reviewer #2: **Yes: ** Habtamu Fekadu Gemede (PhD)

Reviewer #3: No

---

## [Author Response · Author response to Decision Letter 1]

9 Jan 2025

We would like to thank the Editor and the three reviewers for their valuable and detailed comments and suggestions. Please find below our point-by-point response to each of these comments/suggestions. Please note that the line numbers we are referring to in our responses all relate to the manuscript revised version.

Journal Requirements:

and

Response:

The manuscript has been modified according to the journal’s guideline.

2. Thank you for stating the following financial disclosure: "Canadian Institutes of Health Research, Pan American Health Organization and Resolve to Save Lifes."

Response:

The funders had no role in study design, data collection and analysis, decision to publish, or preparation of the manuscript. The funding information has been removed from the manuscript according to requirement.

Response:

All relevant data are within the manuscript and its Supporting Information files. The

branded prepackaged food composition database used in this study is from the

University of Toronto’s Food Label Information Program for Latin America and

Caribbean countries (FLIP-LAC) and FLIP Canada. Any requests to access the FLIP

database can be directed to mary.labbe@utoronto.ca.

4. We note you have included a table to which you do not refer in the text of your manuscript. Please ensure that you refer to Table 5 in your text; if accepted, production will need this reference to link the reader to the Table.

Response:

We have re-named Table 5 as Supporting information S2 Table 2 and included the following line:

“A further breakdown by food categories is presented in Supporting information S2 Table 2.” (line 200-201).

Please note that the 2022 values have been changed as the previous ‘regional’ summary, by mistake, included all 4 Latin America Countries. Panama, which had no longitudinal collection is now removed from the summary for 2022.

Reviewers' comments:

Reviewer #1:

The manuscript is well structured, and the appendices and supplementary files support the findings. However, there are certain things the manuscript need to improve.

1. For the three countries where longitudinal data is available please include results for the test of significance as well if possible.

Response:

Thank you for the suggestion. We have performed Chi-Square test to compare the proportion between 2015-2016 and 2022 to show statistical significance for the overall trend. The p-value has been included in the Fig 2, Fig 3 and Supplementary Table 1.

We have added the following in line 131-133:

“Comparisons between the 2015-2016 and 2022 were analyzed using a Chi-Square test, or Fisher’s exact test for cells with <5 counts.”

2. Add a few paragraphs to support your argument that the actual sodium concentration in the packaged foods are not different from as mentioned on the nutrition facts labels in the results section. Or mention the validation process that were used to verify how accurately the labels represent true value.

Response:

Thank you for highlighting this important limitation. Validation would require laboratory analysis of all the food products included in the study, which is beyond the scope of the current analysis. However, we acknowledge this as a limitation. Therefore, we rely on the sodium values declared by the food companies, and the accuracy depends on their practices. This has been noted in lines 338-344 of the manuscript.

“Furthermore, we were unable to conduct laboratory analyses to validate the sodium concentration in all food products, as this was beyond the scope of the current analysis. Instead we relied on the sodium values reported by food companies, as extracted from the Nutrition facts table (NFt). Although Argentina and Canada have a permitted variation margin of 20% when reporting nutrient values in NFts, the accuracy could not be validated in this study.”

2. The authors have mentioned Atwater validation in the last paragraph of discussion line-288. Please explain how its related to validating sodium concentration, if the validation was used, please provide results accordingly.

The Atwater validation followed the general protocol for checking the quality of nutritional information. It compared the total energy reported in the NFt with the sum of energy provided by each macronutrient, using Atwater’s constants (Carbohydrate: 4kcal/g; Protein: 4kcal/g; Fat: 9kcal/g). This validation was applied to products that declared all three macronutrients and energy (noting that in some countries, the declaration of nutritional information may not be mandatory). However, we acknowledge that this process could not validate sodium content, as sodium does not contribute to energy values. We have revised the paragraph (lines 350-352) and mentioned the limitation that we were not able to validate the actual sodium concentration of foods (line 338-344).

Reviewer #2:

Dear Author, Thank you for your excellent work. I found the detail and really enjoyed in each section of the work. The title is well descriptive, and abstract is well written. Introduction, Methodology and results are very well written. My concern is the discussion section is very shallow and please discuss in detail by comparing with the previous published similar study elsewhere. Therefore I suggested for minor revision.

Response:

We appreciate the reviewers’ comments. For the conciseness of our manuscript, we have not included too many other studies but rather focused on examining the relevant policies in each of the five countries. We have included a few similar studies of other countries:

Line 231-236:

“…This finding is consistent with a previous regional study including 14 LACs, which reported a similar median sodium level for ‘processed meats’ (870mg/100g). Similarly, another regional study with 4 LAC also showed that the highest sodium level was reported in bouillon cubes and powders, meat and fish seasonings and cured and preserved meats…”

Line 318-325:

“… Worth noting that mandatory policies have shown to be more effective than voluntary initiatives. For instance, a recent evaluation in South Africa, one of the fewest countries with mandatory sodium levels for key food categories, showed a reduction in salt intake of 1.15 g/day during the first phase of the program (2015-2019), a significant reduction in population-level sodium intake, and achieved 75% compliance following the second, more stringent phase of legislation in 2019, while other voluntary initiatives in countries like Brazil and Canada have shown only modest results in meeting sodium reduction targets…”

Reviewer #3:

General comment: The manuscript presents important data on sodium content in packaged foods but faces several scientific, writing, and methodological issues. Addressing these problems can significantly enhance the manuscript's clarity, reliability, and overall impact.

Specific comments:

- We don't know why some countries (like Peru and Panama) have different compliance rates. The study's conclusions may be oversimplified without examining socio-economic, regulatory and cultural differences.

Thank you for the comment. We acknowledge that this study focused on reporting the current sodium compliance rate and overall trends observed in some Latin American Countries. Going into depth of examining the differences across countries is beyond the scope of our study, as we are not able to draw clear conclusions between the current policies in each country and the observed compliance rates. However, we have expanded the discussion on policies in each country to provide potential explanations for the findings (line 271-312):

“…For example, Argentina is one of the few countries in the world to established legal sodium maximum levels for certain food groups by passing Act 26905 in 2014, as well as educational campaigns and restaurant policies to reduce sodium intake…

…While Costa Rica only has a voluntary sodium reduction strategy to reduce sodium levels in some key food categories, progress has been monitored continuously. From 2013 to 2021, Costa Rica has a national plan for the reduction of sodium intake, and a private-public partnership was established in 2014 for voluntary national sodium targets…

In the case of Peru, there is no specific national sodium reduction strategy in place; however, Peru implemented ‘high in’ FOPL regulations in 2019 that requires foods exceeding established thresholds for nutrients of concern (i.e., sugar, saturated fat, trans fat, and sodium) to display a ‘high in’ FOPL…

…Canada published a voluntary sodium reduction guideline for prepackaged foods in 2012 and updated the targets in 2020. Evaluations of this initiative have shown minimal progress over time, with only 14% of food categories meeting sodium reduction targets in 2017. In June 2022, Canada introduced its mandatory ‘high in’ FOPL regulations for saturated fat, sodium, and sugar in packaged foods...

…On the other hand, Panama does not have a FOPL policy in place for products with critical excess nutrients, including sodium, which may explain, partially, the low compliance rate with regional SRTs..”.

- There is sampling bias. The study relies on food labels collected from supermarkets, which may not be representative of all packaged foods available in the region.

Thank you for raising this important point. To ensure a comprehensive cohort, we selected the main supermarket chains in each country as representative samples. However, we acknowledge that it was not possible to capture all packaged foods available in the region, and this limitation has been addressed in the limitation section (lines 334-336):

“Additionally, while we selected packaged products from major supermarket chains, we recognize that we may not have captured all of the available packaged food products in the region, or fully represent all components of the population’s diet.”

- The study uses both cross-sectional data from 2022 and longitudinal data from 2015-2022. However, the rationale for combining these two approaches is not fully justified, which may affect the consistency and comparability of the results.

Thank you for the observation. PAHO updated the regional sodium targets in 2021, and products were categorized under the new food categories and assessed based on the new targets. However, we also wanted to report progress in relation to the previous regional targets (2015), as longitudinal monitoring of sodium levels is crucial for understanding trends and informing future policy development. This progress will help determine whether further modifications to the current sodium targets are necessary.

- The study relies on food label information, which may not always accurately reflect the actual sodium content due to possible differences in labelling practices. This limitation is not adequately addressed.

Thank you for pointing out this important limitation. Validation of sodium levels would require a laboratory analysis of the food, which was beyond the scope of our study. Additionally, not all countries have the laboratory capacity to conduct such analyses.

We have added the following paragraph in line 338-344:

“Furthermore, we were unable to conduct laboratory analyses to validate the sodium concentration in all food products, as this was beyond the scope of the current analysis. Instead we relied on the sodium values reported by food companies, as extracted from the Nutrition facts tables (NFt). Although Argentina and Canada have a permitted variation margin of 20% when reporting nutrient values in NFts, the accuracy could not be validated in this study”

- The manuscript does not provide details on how supermarkets were selected for food label collection, raising concerns about potential selection bias.

Thank you for the comments. It is important to highlight the sample selection process to ensure a representative profile of packaged foods from each country. The sample selection was as follows:

1. Argentina: data were collected in Buenos Aires at two of the main supermarket chains. The selected stores are among the six leading retailers in Argentina, which together represent 80% of the grocery retail market.

2. Costa Rica: Supermarkets were selected based on proximity to the research center (INCIENSA). They are part of the two supermarket chains with the highest sales in urban area of Costa Rica, where 70% of the population is concentrated.

3. Panama: Two supermarkets in Panama City from major supermarket chains were selected, located in different neighborhoods, targeting both low-income and high-income groups.

4. Peru: Data collection took place at one of the largest supermarket chains in Peru, which operates the three stores included in this analysis, each targeting to a different socioeconomic sector.

5. Canada: The University of Toronto’s Food Label Information and Price (FLIP) 2020 database was used to obtain data from e-grocery platforms of seven major Canadian retailers, representing over 80% of the grocery retail market in Canada.

For clarity and conciseness, we have added the following paragraph (line 106-108):

“Foods were selected from one or more of the major supermarket chains from different socioeconomic in each country, representing a comprehensive sample of packaged foods across different socioeconomic groups.”

- The manuscript does not address how missing data were handled, which can be a critical issue in longitudinal studies.

Thank you for pointing out this limitation of our study. Missing data were excluded from the analysis, and we did not have data for matching products. Therefore, we could not fully determine whether the overall trend observed in the study was due to the reformation of existing products, the removal of high-sodium products, or the addition of lower-sodium products. This limitation has been added to lines 346-349:

“Regional comparison at different t

---

## [Decision Letter · Decision Letter 1]

23 Jan 2025

PONE-D-24-19881R1Monitoring sodium content in packaged foods sold in the Americas and compliance with the Updated Regional Sodium Reduction TargetsPLOS ONE

Dear Dr. L’Abbé,

Thank you for submitting your manuscript to PLOS ONE. After careful consideration, we feel that it has merit but does not fully meet PLOS ONE’s publication criteria as it currently stands. Therefore, we invite you to submit a revised version of the manuscript that addresses the points raised during the review process.

We look forward to receiving your revised manuscript.

Kind regards,

Trung Quang Nguyen

Academic Editor

PLOS ONE

Journal Requirements:

Reviewers' comments:

Reviewer's Responses to Questions

**Comments to the Author**

1. If the authors have adequately addressed your comments raised in a previous round of review and you feel that this manuscript is now acceptable for publication, you may indicate that here to bypass the “Comments to the Author” section, enter your conflict of interest statement in the “Confidential to Editor” section, and submit your "Accept" recommendation.

Reviewer #1: All comments have been addressed

2. Is the manuscript technically sound, and do the data support the conclusions?

Reviewer #1: Yes

3. Has the statistical analysis been performed appropriately and rigorously? 

Reviewer #1: Yes

4. Have the authors made all data underlying the findings in their manuscript fully available?

Reviewer #1: Yes

5. Is the manuscript presented in an intelligible fashion and written in standard English?

Reviewer #1: Yes

6. Review Comments to the Author

Reviewer #1: The revisions made have effectively addressed the comments raised during the review process, and we appreciate the effort put into improving the manuscript. However, we noticed that Table 1 and Table 2 have not reported the unit of sodium chloride evaluated in the study. It is important to specify whether the measurements are in grams (g) or milligrams (mg) of NaCl per 100g or 100ml of food to ensure clarity and consistency in the presented data. Please update these tables accordingly.

Thank you for your attention to these details.

7. PLOS authors have the option to publish the peer review history of their article (what does this mean? ). If published, this will include your full peer review and any attached files.

**Do you want your identity to be public for this peer review?** For information about this choice, including consent withdrawal, please see our Privacy Policy .

Reviewer #1: No

---

## [Author Response · Author response to Decision Letter 2]

7 Feb 2025

Reviewer #1: The revisions made have effectively addressed the comments raised during the review process, and we appreciate the effort put into improving the manuscript. However, we noticed that Table 1 and Table 2 have not reported the unit of sodium chloride evaluated in the study. It is important to specify whether the measurements are in grams (g) or milligrams (mg) of NaCl per 100g or 100ml of food to ensure clarity and consistency in the presented data. Please update these tables accordingly.

Response:

Thank you for flagging this for us. We have added the units (mg) in Table 1 and Table 2 (line 341-344).

---

## [Decision Letter · Decision Letter 2]

13 Feb 2025

Monitoring sodium content in packaged foods sold in the Americas and compliance with the Updated Regional Sodium Reduction Targets

PONE-D-24-19881R2

Dear Dr. Mary R. L'Abbe,

We’re pleased to inform you that your manuscript has been judged scientifically suitable for publication and will be formally accepted for publication once it meets all outstanding technical requirements.

Kind regards,

Trung Quang Nguyen

Academic Editor

PLOS ONE

Additional Editor Comments (optional):

Reviewers' comments:

Reviewer's Responses to Questions

**Comments to the Author**

1. If the authors have adequately addressed your comments raised in a previous round of review and you feel that this manuscript is now acceptable for publication, you may indicate that here to bypass the “Comments to the Author” section, enter your conflict of interest statement in the “Confidential to Editor” section, and submit your "Accept" recommendation.

Reviewer #1: All comments have been addressed

2. Is the manuscript technically sound, and do the data support the conclusions?

Reviewer #1: Yes

3. Has the statistical analysis been performed appropriately and rigorously? 

Reviewer #1: Yes

4. Have the authors made all data underlying the findings in their manuscript fully available?

Reviewer #1: Yes

5. Is the manuscript presented in an intelligible fashion and written in standard English?

Reviewer #1: Yes

6. Review Comments to the Author

Reviewer #1: All comments have been addressed. I would like to congratulate the authors for their hard work they have put in.

7. PLOS authors have the option to publish the peer review history of their article (what does this mean? ). If published, this will include your full peer review and any attached files.

**Do you want your identity to be public for this peer review?** For information about this choice, including consent withdrawal, please see our Privacy Policy .

Reviewer #1: No

---

## [Editor Report · Acceptance letter]

PONE-D-24-19881R2

PLOS ONE

Dear Dr. L’Abbé,

I'm pleased to inform you that your manuscript has been deemed suitable for publication in PLOS ONE. Congratulations! Your manuscript is now being handed over to our production team.

Kind regards,

on behalf of

Dr. Trung Quang Nguyen

Academic Editor

PLOS ONE